# A Probabilistic Programming Approach To Probabilistic Data Analysis

**Feras Saad**
MIT Probabilistic Computing Project
fsaad@mit.edu

**Vikash Mansinghka**
MIT Probabilistic Computing Project
vkm@mit.edu

## Abstract

Probabilistic techniques are central to data analysis, but different approaches can be challenging to apply, combine, and compare. This paper introduces composable generative population models (CGPMs), a computational abstraction that extends directed graphical models and can be used to describe and compose a broad class of probabilistic data analysis techniques. Examples include discriminative machine learning, hierarchical Bayesian models, multivariate kernel methods, clustering algorithms, and arbitrary probabilistic programs. We demonstrate the integration of CGPMs into BayesDB, a probabilistic programming platform that can express data analysis tasks using a modeling definition language and structured query language. The practical value is illustrated in two ways. First, the paper describes an analysis on a database of Earth satellites, which identifies records that probably violate Kepler's Third Law by composing causal probabilistic programs with non-parametric Bayes in 50 lines of probabilistic code. Second, it reports the lines of code and accuracy of CGPMs compared with baseline solutions from standard machine learning libraries.

## 1  Introduction

Probabilistic techniques are central to data analysis, but can be difficult to apply, combine, and compare. Such difficulties arise because families of approaches such as parametric statistical modeling, machine learning and probabilistic programming are each associated with different formalisms and assumptions. The contributions of this paper are (i) a way to address these challenges by defining CGPMs, a new family of composable probabilistic models; (ii) an integration of this family into BayesDB [10], a probabilistic programming platform for data analysis; and (iii) empirical illustrations of the efficacy of the framework for analyzing a real-world database of Earth satellites.

We introduce composable generative population models (CGPMs), a computational formalism that generalizes directed graphical models. CGPMs specify a table of observable random variables with a finite number of columns and countably infinitely many rows. They support complex intra-row dependencies among the observables, as well as inter-row dependencies among a field of latent random variables. CGPMs are described by a computational interface for generating samples and evaluating densities for random variables derived from the base table by conditioning and marginalization. This paper shows how to package discriminative statistical learning techniques, dimensionality reduction methods, arbitrary probabilistic programs, and their combinations, as CGPMs. We also describe algorithms and illustrate new syntaxes in the probabilistic Metamodeling Language for building composite CGPMs that can interoperate with BayesDB.

The practical value is illustrated in two ways. First, we describe a 50-line analysis that identifies satellite data records that probably violate their theoretical orbital characteristics. The BayesDB script builds models that combine non-parametric Bayesian structure learning with a causal probabilistic program that implements a stochastic variant of Kepler's Third Law. Second, we illustrate coverage and conciseness of the CGPM abstraction by quantifying the improvement in accuracy and reduction in lines of code achieved on a representative data analysis task.

## 2   Composable Generative Population Models

A composable generative population model represents a data generating process for an exchangeable sequence of random vectors $(\boldsymbol{x}_1, \boldsymbol{x}_2, \dots )$, called a population. Each member $\boldsymbol{x}_r$ is $T$-dimensional, and element $x_{[r,t]}$ takes values in an observation space $\mathcal{X}_t$, for $t \in [T]$ and $r \in \mathbb{N}$. A CGPM $\mathcal{G}$ is formally represented by a collection of variables that characterize the data generating process:

$$\mathcal{G} = (\boldsymbol{\alpha}, \boldsymbol{\theta}, \boldsymbol{Z} = \{\boldsymbol{z}_r : r \in \mathbb{N}\}, \boldsymbol{X} = \{\boldsymbol{x}_r : r \in \mathbb{N}\}, \boldsymbol{Y} = \{\boldsymbol{y}_r : r \in \mathbb{N}\}).$$

- $\boldsymbol{\alpha}$: Known, fixed quantities about the population, such as metadata and hyperparameters.

- $\boldsymbol{\theta}$: Population-level latent variables relevant to all members of the population.

- $\boldsymbol{z}_r = (z_{[r,1]}, \dots z_{[r,L]})$: Member-specific latent variables that govern only member $r$ directly.

- $\boldsymbol{x}_r = (x_{[r,1]}, \dots x_{[r,T]})$: Observable output variables for member $r$. A subset of these variables may be observed and recorded in a dataset $\mathcal{D}$.

- $\boldsymbol{y}_r = (y_{[r,1]}, \dots y_{[r,I]})$: Input variables, such as "feature vectors" in a purely discriminative model.

A CGPM is required to satisfy the following conditional independence constraint:

$$\forall r \neq r' \in \mathbb{N}, \forall t, t' \in [T] : x_{[r,t]} \perp\!\!\!\perp x_{[r',t']} \mid \{\boldsymbol{\alpha}, \boldsymbol{\theta}, \boldsymbol{z}_r, \boldsymbol{z}_{r'}\}. \tag{1}$$

Eq (1) formalizes the notion that all dependencies *across* members $r \in \mathbb{N}$ are completely mediated by the population parameters $\boldsymbol{\theta}$ and member-specific variables $\boldsymbol{z}_r$. However, elements $x_{[r,i]}$ and $x_{[r,j]}$ within a member are generally free to assume any dependence structure. Similarly, the member-specific latents in $\boldsymbol{Z}$ may be either uncoupled or highly-coupled given population parameters $\boldsymbol{\theta}$. CGPMs differ from the standard mathematical definition of a joint density in that they are defined in terms of a computational interface (Listing 1). As computational objects, they explicitly distinguish between the *sampler* for the random variables from their joint distribution, and the *assessor* of their joint density. In particular, a CGPM is required to sample/assess the joint distribution of a subset of output variables $\boldsymbol{x}_{[r,Q]}$ conditioned on another subset $\boldsymbol{x}_{[r,E]}$, and marginalizing over $\boldsymbol{x}_{[r,[T]\setminus(Q\cup E)]}$.

---

**Listing 1** Computational interface for composable generative population models.

- $\mathtt{s} \leftarrow \mathtt{simulate}\,(\mathcal{G}, \mathtt{member}: r, \mathtt{query}: Q = \{q_k\}, \mathtt{evidence}: \boldsymbol{x}_{[r,E]}, \mathtt{input}: \boldsymbol{y}_r)$
  Generate a sample from the distribution $\qquad\qquad\qquad\qquad \mathtt{s} \sim^{\mathcal{G}} \boldsymbol{x}_{[r,Q]} | \{\boldsymbol{x}_{[r,E]}, \boldsymbol{y}_r, \mathcal{D}\}.$
- $c \leftarrow \mathtt{logpdf}\,(\mathcal{G}, \mathtt{member}: r, \mathtt{query}: \boldsymbol{x}_{[r,Q]}, \mathtt{evidence}: \boldsymbol{x}_{[r,E]}, \mathtt{input}: \boldsymbol{y}_r)$
  Evaluate the log density $\qquad\qquad\qquad\qquad\quad \log p_{\mathcal{G}}(\boldsymbol{x}_{[r,Q]} | \{\boldsymbol{x}_{[r,E]}, \boldsymbol{y}_r, \mathcal{D}\}).$
- $\mathcal{G}' \leftarrow \mathtt{incorporate}\,(\mathcal{G}, \mathtt{measurement}: x_{[r,t]} \text{ or } \boldsymbol{y}_r)$
  Record a measurement $x_{[r,t]} \in \mathcal{X}_t$ (or $\boldsymbol{y}_r$) into the dataset $\mathcal{D}$.
- $\mathcal{G}' \leftarrow \mathtt{unincorporate}\,(\mathcal{G}, \mathtt{member}: r)$
  Eliminate all measurements of input and output variables for member $r$.
- $\mathcal{G}' \leftarrow \mathtt{infer}\,(\mathcal{G}, \mathtt{program}: \mathcal{T})$
  Adjust internal latent state in accordance with the learning procedure specified by program $\mathcal{T}$.

---

### 2.1   Primitive univariate CGPMs and their statistical data types

The statistical data type (Figure 1) of a population variable $x_t$ generated by a CGPM provides a more refined taxonomy than its "observation space" $\mathcal{X}_t$. The (parameterized) support of a statistical type is the set in which samples from $\mathtt{simulate}$ take values. Each statistical type is also associated with a base measure which ensures $\mathtt{logpdf}$ is well-defined. In high-dimensional populations with heterogeneous types, $\mathtt{logpdf}$ is taken against the product measure of these base measures. The statistical type also identifies invariants that the variable maintains. For instance, the values of a $\mathtt{NOMINAL}$ variable are permutation-invariant. Figure 1 shows statistical data types provided by the Metamodeling Language from BayesDB. The final column shows some examples of primitive CGPMs that are compatible with each statistical type; they implement $\mathtt{logpdf}$ directly using univariate probability density functions, and algorithms for $\mathtt{simulate}$ are well known [4]. For $\mathtt{infer}$ their parameters may be fixed, or learned from data using, e.g., maximum likelihood [2, Chapter 7] or Bayesian priors [5]. We refer to an extended version of this paper [14, Section 3] for using these primitives to implement CGPMs for a broad collection of model classes, including non-parametric Bayes, nearest neighbors, PCA, discriminative machine learning, and multivariate kernel methods.

| Statistical Data Type | Parameters | Support | Measure/$\sigma$-Algebra | Primitive CGPM |
|---|---|---|---|---|
| `BINARY` | - | $\{0, 1\}$ | $(\#, 2^{\{0,1\}})$ | `BERNOULLI` |
| `NOMINAL` | symbols: $S$ | $\{0 \ldots S-1\}$ | $(\#, 2^{[S]})$ | `CATEGORICAL` |
| `COUNT/RATE` | base: $b$ | $\{0, \frac{1}{b}, \frac{2}{b}, \ldots\}$ | $(\#, 2^{\mathbb{N}})$ | `POISSON`, `GEOMETRIC` |
| `CYCLIC` | period: $p$ | $(0, p)$ | $(\lambda, \mathcal{B}(\mathbb{R}))$ | `VON-MISES` |
| `MAGNITUDE` | – | $(0, \infty)$ | $(\lambda, \mathcal{B}(\mathbb{R}))$ | `LOGNORMAL`, `EXPON` |
| `NUMERICAL` | – | $(-\infty, \infty)$ | $(\lambda, \mathcal{B}(\mathbb{R}))$ | `NORMAL` |
| `NUMERICAL-RANGED` | low: $l$, high:$h$ | $(l, h) \subset \mathbb{R}$ | $(\lambda, \mathcal{B}(\mathbb{R}))$ | `BETA`, `NORMAL-TRUNC` |

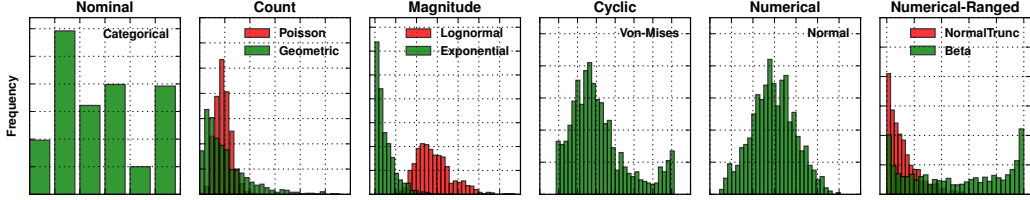

**Figure 1:** Statistical data types for population variables generated by CGPMs available in the BayesDB Metamodeling Language, and samples from their marginal distributions.

## 2.2 Implementing general CGPMs as probabilistic programs in VentureScript

In this section, we show how to implement `simulate` and `logpdf` (Listing 1) for composable generative models written in VentureScript [8], a probabilistic programming language with programmable inference. For simplicity, this section assumes a stronger conditional independence constraint,

$$\exists l, l' \in [L] \text{ such that } (r, t) \neq (r', t') \implies x_{[r,t]} \perp\!\!\!\perp x_{[r',t']} \mid \{\boldsymbol{\alpha}, \boldsymbol{\theta}, z_{[r,l]}, z_{[r',l']}, \boldsymbol{y}_r, \boldsymbol{y}'_r\}. \quad (2)$$

In words, for every observable element $x_{[r,t]}$, there exists a latent variable $z_{[r,l]}$ which (in addition to $\boldsymbol{\theta}$) mediates all coupling with other variables in the population. The member latents $\boldsymbol{Z}$ may still exhibit arbitrary dependencies. The approach for `simulate` and `logpdf` described below is based on approximate inference in tagged subparts of the Venture trace, which carries a full realization of all random choices (population and member-specific latent variables) made by the program. The runtime system carries a set of $K$ traces $\{(\boldsymbol{\theta}^k, \boldsymbol{Z}^k)\}_{k=1}^{K}$ sampled from an approximate posterior $p_{\mathcal{G}}(\boldsymbol{\theta}, \boldsymbol{Z}|\mathcal{D})$. These traces are assigned weights depending on the user-specified evidence $\boldsymbol{x}_{[r,E]}$ in the `simulate`/`logpdf` function call. $\mathcal{G}$ represents the CGPM as a probabilistic program, and the input $\boldsymbol{y}_r$ and latent variables $\boldsymbol{Z}^k$ are treated as ambient quantities in $\boldsymbol{\theta}^k$. The distribution of interest is

$$p_{\mathcal{G}}(\boldsymbol{x}_{[r,Q]}|\boldsymbol{x}_{[r,E]}, \mathcal{D}) = \int_{\boldsymbol{\theta}} p_{\mathcal{G}}(\boldsymbol{x}_{[r,Q]}|\boldsymbol{x}_{[r,E]}, \boldsymbol{\theta}, \mathcal{D}) p_{\mathcal{G}}(\boldsymbol{\theta}|\boldsymbol{x}_{[r,E]}, \mathcal{D}) d\boldsymbol{\theta}$$

$$= \int_{\boldsymbol{\theta}} p_{\mathcal{G}}(\boldsymbol{x}_{[r,Q]}|\boldsymbol{x}_{[r,E]}, \boldsymbol{\theta}, \mathcal{D}) \left( \frac{p_{\mathcal{G}}(\boldsymbol{x}_{[r,E]}|\boldsymbol{\theta}, \mathcal{D}) p_{\mathcal{G}}(\boldsymbol{\theta}|\mathcal{D})}{p_{\mathcal{G}}(\boldsymbol{x}_{[r,E]}|\mathcal{D})} \right) d\boldsymbol{\theta} \quad (3)$$

$$\approx \frac{1}{\sum_{k=1}^{K} w^k} \sum_{k=1}^{K} p_{\mathcal{G}}(\boldsymbol{x}_{[r,Q]}|\boldsymbol{x}_{[r,E]}, \boldsymbol{\theta}^k, \mathcal{D}) w^k \qquad \text{where } \boldsymbol{\theta}^k \sim^{\mathcal{G}} |\mathcal{D}. \quad (4)$$

The weight $w^k = p_{\mathcal{G}}(\boldsymbol{x}_{[r,E]}|\boldsymbol{\theta}^k, \mathcal{D})$ of trace $\boldsymbol{\theta}^k$ is the likelihood of the evidence. The weighting scheme (4) is a computational trade-off avoiding the requirement to run posterior inference on population parameters $\boldsymbol{\theta}$ for a query about member $r$. It suffices to derive the distribution for only $\boldsymbol{\theta}^k$,

$$p_{\mathcal{G}}(\boldsymbol{x}_{[r,Q]}|\boldsymbol{x}_{[r,E]}, \boldsymbol{\theta}^k, \mathcal{D}) = \int_{\boldsymbol{z}_r^k} p_{\mathcal{G}}(\boldsymbol{x}_{[r,Q]}, \boldsymbol{z}_r^k|\boldsymbol{x}_{[r,E]}, \boldsymbol{\theta}^k, \mathcal{D}) d\boldsymbol{z}_r^k \quad (5)$$

$$= \int_{\boldsymbol{z}_r^k} \prod_{q \in Q} \left( p_{\mathcal{G}}(x_{[r,q]}|\boldsymbol{z}_r^k, \boldsymbol{\theta}^k) \right) p_{\mathcal{G}}(\boldsymbol{z}_r^k|\boldsymbol{x}_{[r,E]}, \boldsymbol{\theta}^k, \mathcal{D}) d\boldsymbol{z}_r^k \approx \frac{1}{J} \sum_{j=1}^{J} \prod_{q \in Q} p_{\mathcal{G}}(x_{[r,q]}|\boldsymbol{z}_r^{k,j}, \boldsymbol{\theta}^k), \quad (6)$$

where $\boldsymbol{z}_r^{k,j} \sim^{\mathcal{G}} |\{\boldsymbol{x}_{[r,E]}, \boldsymbol{\theta}^k, \mathcal{D}\}$. Eq (5) suggests that `simulate` can be implemented by sampling $(\boldsymbol{x}_{[r,Q]}, \boldsymbol{z}_r^k) \sim^{\mathcal{G}} |\{\boldsymbol{x}_{[r,E]}, \boldsymbol{\theta}^k, \mathcal{D}\}$ from the joint local posterior, then returning elements $\boldsymbol{x}_{[r,Q]}$. Eq (6) shows that `logpdf` can be implemented by first sampling the member latents $\boldsymbol{z}_r^k \sim^{\mathcal{G}} |\{\boldsymbol{x}_{[r,E]}, \boldsymbol{\theta}^k, \mathcal{D}\}$ from the local posterior; using the conditional independence constraint (2), the query $\boldsymbol{x}_{[r,Q]}$ then factors into a product of density terms for each element $x_{[r,q]}$.

To aggregate over $\{\boldsymbol{\theta}^k\}_{k=1}^K$, for `simulate` the runtime obtains the queried sample by first drawing $k \sim \text{CATEGORICAL}(\{w^1, \dots, w^K\})$, then returns the sample $\boldsymbol{x}_{[r,Q]}$ drawn from trace $\boldsymbol{\theta}^k$. Similarly, `logpdf` is computed using the weighted Monte Carlo estimator (6). Algorithms 2a and 2b summarize implementations of `simulate` and `logpdf` in a general probabilistic programming environment.

---

**Algorithm 2a** `simulate` for CGPMs in a probabilistic programming environment.

---

1: **function** SIMULATE($\mathcal{G}, r, Q, \boldsymbol{x}_{[r,E]}, \boldsymbol{y}_r$)
2:   **for** $k = 1, \dots, K$ **do**                                                  $\triangleright$ for each trace $k$
3:     **if** $\boldsymbol{z}_r^k \notin \boldsymbol{Z}^k$ **then**                               $\triangleright$ if member $r$ has unknown local latents
4:       $\boldsymbol{z}_r^k \sim^{\mathcal{G}} |\{\boldsymbol{\theta}^k, \boldsymbol{Z}^k, \mathcal{D}\}$                $\triangleright$ sample them from the prior
5:     $w^k \leftarrow \prod_{e \in E} p_{\mathcal{G}}(x_{[r,e]}|\boldsymbol{\theta}^k, \boldsymbol{z}_r^k)$       $\triangleright$ weight the trace by likelihood of evidence
6:   $k \sim \text{CATEGORICAL}(\{w^1, \dots, w^k\})$                      $\triangleright$ importance resample the traces
7:   $\{\boldsymbol{x}_{[r,Q]}, \boldsymbol{z}_r^k\} \sim^{\mathcal{G}} |\{\boldsymbol{\theta}^k, \boldsymbol{Z}^k, \mathcal{D} \cup \{\boldsymbol{y}_r, \boldsymbol{x}_{[r,E]}\}\}$   $\triangleright$ run a transition operator leaving target invariant
8:   **return** $\boldsymbol{x}_{[r,Q]}$                                          $\triangleright$ select query variables from the resampled trace

---

**Algorithm 2b** `logpdf` for CGPMs in a probabilistic programming environment.

---

1: **function** LOGPDF($\mathcal{G}, r, \boldsymbol{x}_{[r,Q]}, \boldsymbol{x}_{[r,E]}, \boldsymbol{y}_r$)
2:   **for** $k = 1, \dots, K$ **do**                                                  $\triangleright$ for each trace $k$
3:     Run steps 2 through 5 from Algorithm 2a                         $\triangleright$ retrieve the trace weight
4:     **for** $j = 1, \dots, J$ **do**                                     $\triangleright$ obtain $J$ samples of latents in scope of member $r$
5:       $\boldsymbol{z}_r^{k,j} \sim^{\mathcal{G}} |\{\boldsymbol{\theta}^k, \boldsymbol{Z}^k, \mathcal{D} \cup \{\boldsymbol{y}_r, \boldsymbol{x}_{[r,E]}\}\}$   $\triangleright$ run a transition operator leaving target invariant
6:       $h^{k,j} \leftarrow \prod_{q \in Q} p_{\mathcal{G}}(x_{[r,q]}|\boldsymbol{\theta}^k, \boldsymbol{z}_r^{k,j})$          $\triangleright$ compute the density estimate
7:     $r^k \leftarrow \frac{1}{J} \sum_{j=1}^J h^{k,j}$                      $\triangleright$ aggregate density estimates by simple Monte Carlo
8:     $q^k \leftarrow r^k w^k$                                            $\triangleright$ importance weight the estimate
9:   **return** $\log\left(\sum_{k=1}^K q^k\right) - \log\left(\sum_{k=1}^K w^k\right)$           $\triangleright$ weighted importance sampling over all traces

---

## 2.3 Inference in a composite network of CGPMs

This section shows how CGPMs are composed by applying the output of one to the input of another. This allows us to build complex probabilistic models out of simpler primitives directly as software. Section 3 demonstrates surface-level syntaxes in the Metamodeling Language for constructing these composite structures. We report experiments including up to three layers of composed CGPMs.

Let $\mathcal{G}^a$ be a CGPM with output $\boldsymbol{x}_*^a$ and input $\boldsymbol{y}_*^a$, and $\mathcal{G}^b$ have output $\boldsymbol{x}_*^b$ and input $\boldsymbol{y}_*^b$ (the symbol $*$ indexes all members $r \in \mathbb{N}$). The composition $\mathcal{G}_{\mathcal{B}}^b \circ \mathcal{G}_{\mathcal{A}}^a$ applies the subset of outputs $\boldsymbol{x}_{[*,\mathcal{A}]}^a$ of $\mathcal{G}^a$ to the inputs $\boldsymbol{y}_{[*,\mathcal{B}]}^b$ of $\mathcal{G}^b$, where $|\mathcal{A}| = |\mathcal{B}|$ and the variables are type-matched (Figure 1). This operation results in a new CGPM $\mathcal{G}^c$ with output $\boldsymbol{x}_*^a \cup \boldsymbol{x}_*^b$ and input $\boldsymbol{y}_*^a \cup \boldsymbol{y}_{[*,\backslash\mathcal{B}]}^b$. In general, a collection $\{\mathcal{G}^k : k \in [K]\}$ of CGPMs can be organized into a generalized directed graph $\mathcal{G}^{[K]}$, which itself is a CGPM. Node $k$ is an "internal" CGPM $\mathcal{G}^k$, and the labeled edge $a_{\mathcal{A}} \to b_{\mathcal{B}}$ denotes the composition $\mathcal{G}_{\mathcal{A}}^a \circ \mathcal{G}_{\mathcal{B}}^b$. The directed acyclic edge structure applies only to edges between elements of different CGPMs in the network; elements $x_{[*,i]}^k, x_{[*,j]}^k$ within $\mathcal{G}^k$ may satisfy the more general constraint (1).

Algorithms 3a and 3b show sampling-importance-resampling and ratio-likelihood weighting algorithms that combine `simulate` and `logpdf` from each individual $\mathcal{G}^k$ to compute queries against network $\mathcal{G}^{[K]}$. The symbol $\pi^k = \{(p,t) : x_{[*,t]}^p \in \boldsymbol{y}_*^k\}$ refers to the set of all output elements from upstream CGPMs connected to the inputs of $\mathcal{G}^k$, so that $\{\pi^k : k \in [K]\}$ encodes the graph adjacency matrix. Subroutine 3c generates a full realization of all unconstrained variables, and weights forward samples from the network by the likelihood of constraints. Algorithm 3b is based on ratio-likelihood weighting (both terms in line 6 are computed by unnormalized importance sampling) and admits an analysis with known error bounds when `logpdf` and `simulate` of each $\mathcal{G}^k$ are exact [7].

---

**Algorithm 3a** `simulate` in a directed acyclic network of CGPMs.

---

1: **function** SIMULATE($\mathcal{G}^k, r, Q^k, \boldsymbol{x}_{[r,E^k]}^k, \boldsymbol{y}_r^k$, for $k \in [K]$)
2:   **for** $j = 1, \dots, J$ **do**                                          $\triangleright$ generate $J$ importance samples
3:     $(\boldsymbol{s}^j, w^j) \leftarrow$ WEIGHTED-SAMPLE $(\{\boldsymbol{x}_{[r,E^k]}^k : k \in [K]\})$   $\triangleright$ retrieve $j$th weighted sample
4:   $m \leftarrow \text{CATEGORICAL}(\{w^1, \dots, w^J\})$               $\triangleright$ resample by importance weights
5:   **return** $\{\boldsymbol{x}_{[r,Q^k]}^k \in \boldsymbol{s}_m : k \in [K]\}$             $\triangleright$ return query variables from the selected sample

---

**Algorithm 3b** `logpdf` in a directed acyclic network of CGPMs.

1: **function** SIMULATE($\mathcal{G}^k, r, \boldsymbol{x}_Q^k, \boldsymbol{x}_{[r,E^k]}^k, \boldsymbol{y}_r^k$, for $k \in [K]$)
2:      **for** $j = 1, \ldots, J$ **do**                ▷ generate $J$ importance samples
3:          $(\boldsymbol{s}^j, w^j) \leftarrow$ WEIGHTED-SAMPLE $(\{\boldsymbol{x}_{[r,Q^k \cup E^k]}^k : k \in [K]\})$    ▷ joint density of query/evidence
4:      **for** $j = 1, \ldots, J'$ **do**              ▷ generate $J'$ importance samples
5:          $(\boldsymbol{s}'^j, w'^j) \leftarrow$ WEIGHTED-SAMPLE $(\{\boldsymbol{x}_{[r,E^k]}^k : k \in [K]\})$    ▷ marginal density of evidence
6:      **return** $\log\left(\sum_{[J]} w^j / \sum_{[J']} w'^j\right) - \log(J/J')$     ▷ return likelihood ratio importance estimate

**Algorithm 3c** Weighted forward sampling in a directed acyclic network of CGPMs.

1: **function** WEIGHTED-SAMPLE (constraints: $\boldsymbol{x}_{[r,C^k]}^k$, for $k \in [K]$)
2:      $(\boldsymbol{s}, \log w) \leftarrow (\varnothing, 0)$                  ▷ initialize empty sample with zero weight
3:      **for** $k \in$ TOPOSORT $(\{\pi^1, \ldots, \pi^K\})$ **do**       ▷ topologically sort CGPMs using adjacency matrix
4:          $\tilde{\boldsymbol{y}}_r^k \leftarrow \boldsymbol{y}_r^k \cup \{x_{[r,t]}^p \in \boldsymbol{s} : (p,t) \in \pi^k\}$       ▷ retrieve required inputs at node $k$
5:          $\log w \leftarrow \log w + \text{logpdf} (\mathcal{G}^k, r, \boldsymbol{x}_{[r,C^k]}^k, \varnothing, \tilde{\boldsymbol{y}}_r^k)$    ▷ update weight by likelihood of constraint
6:          $\boldsymbol{x}_{[r,\backslash C^k]}^k \leftarrow \text{simulate} (\mathcal{G}^k, r, \backslash C^k, \boldsymbol{x}_{[r,C^k]}^k, \tilde{\boldsymbol{y}}_r^k)$    ▷ simulate unconstrained nodes
7:          $\boldsymbol{s} \leftarrow \boldsymbol{s} \cup \boldsymbol{x}_{[r,C^k \cup \backslash C^k]}^k$            ▷ append all node values to sample
8:      **return** $(\boldsymbol{s}, w)$                  ▷ return the overall sample and its weight

# 3 Analyzing satellites using CGPMs built from causal probabilistic programs, discriminative machine learning, and Bayesian non-parametrics

This section outlines a case study applying CGPMs to a database of 1163 satellites maintained by the Union of Concerned Scientists [12]. The dataset contains 23 numerical and categorical features of each satellite such as its material, functional, physical, orbital and economic characteristics. The list of variables and examples of three representative satellites are shown in Table 1. A detailed study of this database using BayesDB provided in [10]. Here, we compose the baseline CGPM in BayesDB, CrossCat [9], a non-parametric Bayesian structure learner for high dimensional data tables, with several CGPMs: a classical physics model written in VentureScript, a random forest classifier, factor analysis, and an ordinary least squares regressor. These composite models allow us to identify satellites that probably violate their orbital mechanics (Figure 2), as well as accurately infer the anticipated lifetimes of new satellites (Figure 3). We refer to [14, Section 6] for several more experiments on a broader set of data analysis tasks, as well as comparisons to baseline machine learning solutions.

| | | | |
|---|---|---|---|
| **Name** | International Space Station | AAUSat-3 | Advanced Orion 5 (NRO L-32, USA 223) |
| **Country of Operator** | Multinational | Denmark | USA |
| **Operator Owner** | NASA/Multinational | Aalborg University | National Reconnaissance Office (NRO) |
| **Users** | Government | Civil | Military |
| **Purpose** | Scientific Research | Technology Development | Electronic Surveillance |
| **Class of Orbit** | LEO | LEO | GEO |
| **Type of Orbit** | Intermediate | NaN | NaN |
| **Perigee km** | 401 | 770 | 35500 |
| **Apogee km** | 422 | 787 | 35500 |
| **Eccentricity** | 0.00155 | 0.00119 | 0 |
| **Period minutes** | 92.8 | 100.42 | NaN |
| **Launch Mass kg** | NaN | 0.8 | 5000 |
| **Dry Mass kg** | NaN | NaN | NaN |
| **Power watts** | NaN | NaN | NaN |
| **Date of Launch** | 36119 | 41330 | 40503 |
| **Anticipated Lifetime** | 30 | 1 | NaN |
| **Contractor** | Boeing Satellite Systems/Multinational | Aalborg University | National Reconnaissance Laboratory |
| **Country of Contractor** | Multinational | Denmark | USA |
| **Launch Site** | Baikonur Cosmodrome | Satish Dhawan Space Center | Cape Canaveral |
| **Launch Vehicle** | Proton | PSLV | Delta 4 Heavy |
| **Source Used for Orbital Data** | www.satellitedebris.net 12/12 | SC - ASCR | SC - ASCR |
| **longitude radians of geo** | NaN | NaN | 1.761037215 |
| **Inclination radians** | 0.9005899 | 1.721418241 | 0 |

**Table 1: Variables in the satellite population, and three representative satellites.** The records are multivariate, heterogeneously typed, and contain arbitrary patterns of missing data.

```
1   CREATE TABLE satellites_ucs FROM 'satellites.csv';
2   CREATE POPULATION satellites FOR satellites_ucs WITH SCHEMA ( GUESS STATTYPES FOR (*) );
3
4   CREATE METAMODEL satellites_hybrid FOR satellites WITH BASELINE CROSSCAT (
5
6     OVERRIDE GENERATIVE MODEL FOR type_of_orbit
7     GIVEN apogee_km, perigee_km, period_minutes, users, class_of_orbit
8     USING RANDOM_FOREST (num_categories = 7);
9
10    OVERRIDE GENERATIVE MODEL FOR launch_mass_kg, dry_mass_kg, power_watts, perigee_km, apogee_km
11    USING FACTOR_ANALYSIS (dimensionality = 2);
12
13    OVERRIDE GENERATIVE MODEL FOR period_minutes
14    AND EXPOSE kepler_cluster_id CATEGORICAL, kepler_noise NUMERICAL
15    GIVEN apogee_km, perigee_km USING VENTURESCRIPT (program = '
16      define dpmm_kepler = () -> {                    // Definition of DPMM Kepler model program.
17        assume keplers_law = (apogee, perigee) -> {
18          (GM, earth_radius) = (398600, 6378);
19          a = .5*(abs(apogee) + abs(perigee)) + earth_radius;
20          2 * pi * sqrt(a**3 / GM) / 60 };
21        // Latent variable priors.
22        assume crp_alpha = gamma(1,1);
23        assume cluster_id_sampler = make_crp(crp_alpha);
24        assume noise_sampler = mem((cluster) -> make_nig_normal(1, 1, 1, 1));
25        // Simulator for latent variables (kepler_cluster_id and kepler_noise).
26        assume sim_cluster_id = mem((rowid, apogee, perigee) -> {
27          cluster_id_sampler() #rowid:1 });
28        assume sim_noise = mem((rowid, apogee, perigee) -> {
29          cluster_id = sim_cluster_id(rowid, apogee, perigee);
30          noise_sampler(cluster_id)() #rowid:2 });
31        // Simulator for observable variable (period_minutes).
32        assume sim_period = mem((rowid, apogee, perigee) -> {
33          keplers_law(apogee, perigee) + sim_noise(rowid, apogee, perigee) });
34        assume outputs = [sim_period, sim_cluster_id, sim_noise];     // List of output variables.
35      };
36      // Procedures for observing the output variables.
37      define obs_cluster_id = (rowid, apogee, perigee, value, label) -> {
38        $label: observe sim_cluster_id( $rowid, $apogee, $perigee) = atom(value); };
39      define obs_noise = (rowid, apogee, perigee, value, label) -> {
40        $label: observe sim_noise( $rowid, $apogee, $perigee) = value; };
41      define obs_period = (rowid, apogee, perigee, value, label) -> {
42        theoretical_period = run(sample keplers_law($apogee, $perigee));
43        obs_noise( rowid, apogee, perigee, value - theoretical_period, label); };
44      define observers = [obs_period, obs_cluster_id, obs_noise];    // List of observer procedures.
45      define inputs = ["apogee", "perigee"];                         // List of input variables.
46      define transition = (N) -> { default_markov_chain(N) };        // Transition operator.
47  '));
48  INITIALIZE 10 MODELS FOR satellites_hybrid;
49  ANALYZE satellites_hybrid FOR 100 ITERATIONS;
50  INFER name, apogee_km, perigee_km, period_minutes, kepler_cluster_id, kepler_noise FROM satellites;
```

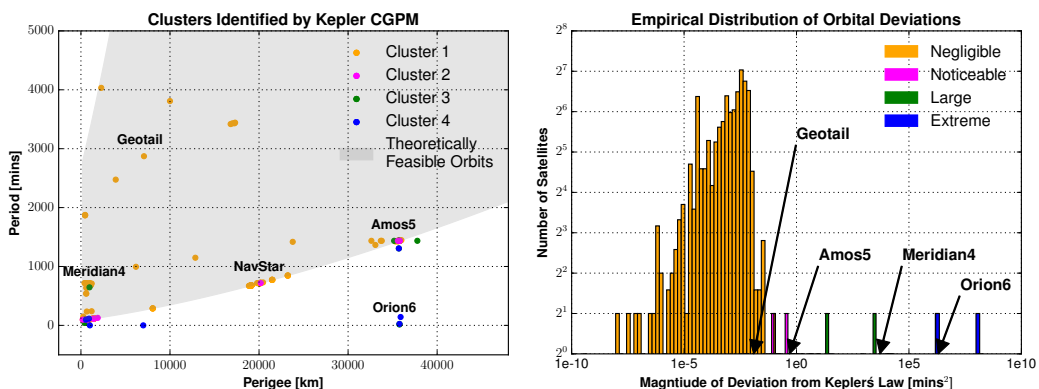

**Figure 2: A session in BayesDB to detect satellites whose orbits are likely violations of Kepler's Third Law using a causal composable generative population model written in VentureScript.** The dpmm_kepler CGPM (line 17) learns a DPMM on the residuals of each satellite's deviation from its theoretical orbit. Both the cluster identity and inferred noise are exposed latent variables (line 14). Each dot in the scatter plot (left) is a satellite in the population, and its color represents the latent cluster assignment learned by dpmm_kepler. The histogram (right) shows that each of the four detected clusters roughly translates to a qualitative description of the deviation: yellow (negligible), magenta (noticeable), green (large), and blue (extreme).

```
1   CREATE TABLE data_train FROM 'sat_train.csv';
2   .nullify data_train 'NaN';
3
4   CREATE POPULATION satellites FOR data_train
5     WITH SCHEMA(
6       GUESS STATTYPES FOR (*)
7   );
8
9   CREATE METAMODEL crosscat_ols FOR satellites
10    WITH BASELINE CROSSCAT(
11      OVERRIDE GENERATIVE MODEL FOR
12          anticipated_lifetime
13      GIVEN
14          type_of_orbit, perigee_km, apogee_km,
15          period_minutes, date_of_launch,
16          launch_mass_kg
17      USING LINEAR_REGRESSION
18  );
19
20  INITIALIZE 4 MODELS FOR crosscat_ols;
21  ANALYZE crosscat_ols FOR 100 ITERATION WAIT;
22
23  CREATE TABLE data_test FROM 'sat_test.csv';
24  .nullify data_test 'NaN';
25  .sql INSERT INTO data_train
26      SELECT * FROM data_test;
27
28  CREATE TABLE predicted_lifetime AS
29      INFER EXPLICIT
30          PREDICT anticipated_lifetime
31          CONFIDENCE prediction_confidence
32      FROM satellites WHERE _rowid_ > 1000;
```

```python
def dummy_code_categoricals(frame, maximum=10):

    def dummy_code_categoricals(series):
        categories = pd.get_dummies(
            series, dummy_na=1)
        if len(categories.columns) > maximum-1:
            return None
        if sum(categories[np.nan]) == 0:
            del categories[np.nan]
        categories.drop(
            categories.columns[-1], axis=1,
            inplace=1)
        return categories

def append_frames(base, right):
    for col in right.columns:
        base[col] = pd.DataFrame(right[col])

numerical = frame.select_dtypes([float])
categorical = frame.select_dtypes([object])

categorical_coded = filter(
    lambda s: s is not None,
    [dummy_code_categoricals(categorical[c])
        for c in categorical.columns])

joined = numerical

for sub_frame in categorical_coded:
    append_frames(joined, sub_frame)

return joined
```

**(a)** Full session in BayesDB which loads the training and test sets, creates a hybrid CGPM, and runs the regression using CrossCat+OLS.

**(b)** Ad-hoc Python routine (used by baselines) for coding nominal predictors in a dataframe with missing values and mixed data types.

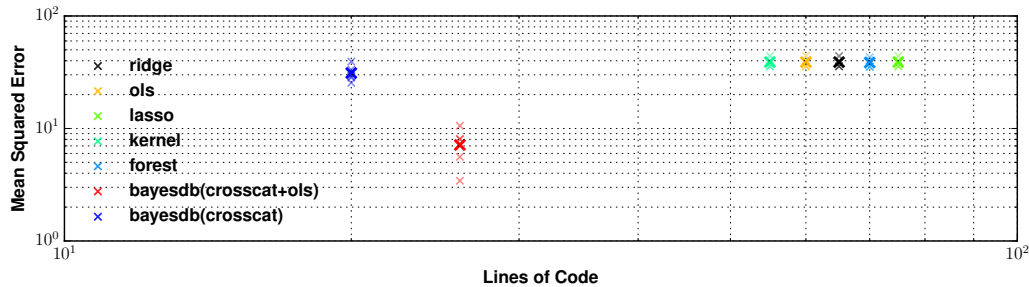

**Figure 3: In a high-dimensional regression problem with mixed data types and missing data, the composite CGPM improves prediction accuracy over purely generative and purely discriminative baselines.** The task is to infer the anticipated lifetime of a held-out satellite given categorical and numerical features such as type of orbit, launch mass, and orbital period. As feature vectors in the test set have missing entries, purely discriminative models (ridge, lasso, OLS) either heuristically impute missing features, or ignore the features and predict the anticipated lifetime using the mean in the training set. The purely generative model (CrossCat) can impute missing features from their joint distribution, but only indirectly mediates dependencies between the predictors and response through latent variables. The composite CGPM (CrossCat+OLS) in panel (a) combines advantages of both approaches; statistical imputation followed by regression on the features leads to improved predictive accuracy. The reduced code size is a result of using SQL, BQL, & MML, for preprocessing, model-building and predictive querying, as opposed to collections of ad-hoc scripts such as panel (b).

Figure 2 shows the MML program for constructing the hybrid CGPM on the satellites population. In terms of the compositional formalism from Section 2.3, the CrossCat CGPM (specified by the MML BASELINE keyword) learns the joint distribution of variables at the "root" of the network (i.e., all variables from Table 1 which do not appear as arguments to an MML OVERRIDE command). The dpmm_kepler CGPM in line 16 of the top panel in Figure 2 accepts apogee_km and perigee_km as input variables $\boldsymbol{y} = (A, P)$, and produces as output the period_minutes $\boldsymbol{x} = (T)$. These variables characterize the elliptical orbit of a satellite and are constrained by the relationships $e = (A - P)/(A + P)$ and $T = 2\pi\sqrt{((A + P)/2))^3/GM}$ where $e$ is the eccentricity and $GM$

is a physical constant. The program specifies a stochastic version of Kepler's Law using a Dirichlet process mixture model for the distribution over errors (between the theoretical and observed period),

$$P \sim \mathrm{DP}(\alpha, \text{Normal-Inverse-Gamma}(m, V, a, b)), \qquad (\mu_r, \sigma_r^2) | P \sim P$$

$$\epsilon_r | \{\mu_r, \sigma_r^2, \boldsymbol{y}_r\} \sim \text{Normal}(\cdot | \mu_r, \sigma_r^2), \text{ where } \epsilon_r := T_r - \text{Kepler}(A_r, P_r).$$

The lower panels of Figure 2 illustrate how the `dpmm_kepler` CGPM clusters satellites based on the magnitude of the deviation from their theoretical orbits; the variables (deviation, cluster identity, etc) in these figures are obtained from the BQL query on line 50. For instance, the satellite `Orion6` shown in the right panel of Figure 2, belongs to a component with "extreme" deviation. Further investigation reveals that `Orion6` has a recorded period 23.94 minutes, most likely a data entry error for the true period of 24 *hours* (1440 minutes); we have reported such errors to the maintainers of the database.

The data analysis task in Figure 3 is to infer the `anticipated_lifetime` $x_r$ of a new satellite, given a set of features $\boldsymbol{y}_r$ such as its `type_of_orbit` and `perigee_km`. A simple OLS regressor with normal errors is used for the response $p_{\mathcal{G}^{\mathrm{ols}}}(x_r | \boldsymbol{y}_r)$. The CrossCat baseline learns a joint generative model for the covariates $p_{\mathcal{G}^{\mathrm{crosscat}}}(\boldsymbol{y}_r)$. The composite CGPM `crosscat_ols` built Figure 3 (left panel) thus carries the full joint distribution over the predictors and response $p_{\mathcal{G}}(\boldsymbol{x}_r, \boldsymbol{y}_r)$, leading to more accurate predictions. Advantages of this hybrid approach are further discussed in the figure.

## 4  Related Work and Discussion

This paper has shown that it is possible to use a computational formalism in probabilistic programming to uniformly apply, combine, and compare a broad class of probabilistic data analysis techniques. By integrating CGPMs into BayesDB [10] and expressing their compositions in the Metamodeling Language, we have shown it is possible to combine CGPMs synthesized by automatic model discovery [9] with custom probabilistic programs, which accept and produce multivariate inputs and outputs, into coherent joint probabilistic models. Advantages of this hybrid approach to modeling and inference include combining the strengths of both generative and discriminative techniques, as well as savings in code complexity from the uniformity of the CGPM interface.

While our experiments have constructed CGPMs using VentureScript and Python implementations, the general probabilistic programming interface of CGPMs makes it possible for BayesDB to interact with a variety systems such as BUGS [15], Stan [1], BLOG [11], Figaro [13], and others. Each of these systems provides varying levels of model expressiveness and inference capabilities, and can be used to be construct domain-specific CGPMs with different performance properties based on the data analysis task on hand. Moreover, by expressing the data analysis tasks in BayesDB using the model-independent Bayesian Query Language [10, Section 3], CGPMs can be queried without necessarily exposing their internal structures to end users. Taken together, these characteristics help illustrate the broad utility of the BayesDB probabilistic programming platform and architecture [14, Section 5], which in principle can be used to create and query novel combinations of black-box machine learning, statistical modeling, computer simulation, and probabilistic generative models.

Our applications have so far focused on CGPMs for analyzing populations from standard multivariate statistics. A promising area for future work is extending the computational abstraction of CGPMs, as well as the Metamodeling and Bayesian Query Languages, to cover analysis tasks in other domains such longitudinal populations [3], statistical relational settings [6], or natural language processing and computer vision. Another extension, important in practice, is developing alternative compositional algorithms for querying CGPMs (Section 2.3). The importance sampling strategy used for compositional `simulate` and `logpdf` may only be feasible when the networks are shallow and the constituent CGPMs are fairly noisy; better Monte Carlo strategies or perhaps even variational strategies may be needed for deeper networks. Additional future work for composite CGPMs include (i) algorithms for jointly learning the internal parameters of each individual CGPM, using, e.g., imputations from its parents, and (ii) new meta-algorithms for structure learning among a collection of compatible CGPMs, in a similar spirit to the non-parametric divide-and-conquer method from [9].

We hope the formalisms in this paper lead to practical, unifying tools for data analysis that integrate these ideas, and provide abstractions that enable the probabilistic programming community to collaboratively explore these research directions.

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
