[Reviews · NeurIPS 2016]

Reviewer 1

Summary

This paper extends GPMs (e.g. [4]) to conditional GPMs, and implements them in Venturescript. I found the description of the expts (sec 4) to be too compressed to be useful. Also I have concerns about whether other prob programming systems (e.g. BUGS, STAN) could be used for these tasks.

Qualitative Assessment

This paper is about conditional generative population models (CPGMs). This enables supervised learning tasks to be tackled. The method is demonstrated from some tasks involving satellite data. Sec 4.1 concerns violations of Kepler's third law, based on the equation T = ... in line 151. Some other results are in Fig 4 concerning other prediction tasks. The presentation of the both sets of results is way too compressed, and is a serious deficiency. It is very hard to follow what is going on here. It looks to me that the paper has become imbalanced between the prob programming presentation (which I guess the authors are most interested in), and a coherent explanation of the demonstrations. The authors should remember that supp mat can be used if necessary. Another concern to me was about competitor prob programming systems. I am not an expert on this topic, but from talks I have seen I would guess that frameworks like BUGS and STAN could handle these kinds of models already, but these are not discussed in sec 5. If that assumption is incorrect the authors will no doubt rebut ... BTW the standard mixture of experts (Jabobs et al, Adaptive Mixtures of Local Experts, N Comp, 1991) allows dependence of p(z) [the gating network] on the associated y's -- is this possible in your framework? It appears not from Fig 1. Overall: as far as I can understand this paper advances the work of [4] by introducing conditonal GPMs, and implementing them in Venturescript. CPGMs are applied to some satellite data, but the detail of these experiments is not well-enough explained. Other points: - l 40 LOC is undefined at this point. - In standard supervised learning the y's are the targets and x's are the inputs. It might help readability to stick to this convention.

Confidence in this Review

2-Confident (read it all; understood it all reasonably well)


Reviewer 2

Summary

This paper shows conditional generative population models as an extension of PGMs. To illustrate this the paper takes the default BayesDB example of satellite orbits and shows how to find errors in the observed data given expected behaviour.

Qualitative Assessment

This paper takes the default BayesDB example of satellite orbits and shows how to find errors in the observed data given expected behaviour. To achieve this, ths authors construct a new type of generative population model and implement this model as part of the BayesDB/VentureScript environment. Overall I like that this pushes for more complex data analysis tasks in a general probabilistic programming environment. The paper, however, is not an easy read and it is unclear whether the proposed extension are really that general and not tuned towards the orbital example. The authors expect deep knowledge about a number of systems (BayesDB, VentureScript, Crosscat) without clearly showing the difference . This makes that it is unclear whether they implemented the CGPM as part of the BayesDB/VentureScript engine or on top of it using only existing language features. Also the example is missing some background. It is clear how the proposed method handles it, and I like the comparison with baseline methods but I am confused why you would solve this with random forests instead of, say, Kalman filters given that such techniques fit better the required mathematics for the physical laws and are a popular method to compute predicted orbits. As a result this one example on which the new model is shown feels pushed into a suboptimal setting. Detailed comments: - The title of the paper is too general. The system this paper expands upon, BayesDB, already uses this concept which makes it unclear what this paper adds. The contents of the paper is how to implement a certain model into VentureScript which then can be used for data analysis. - The paper switches the description of CGPMs as a generalization of directed graphical models and a generalization of generative population models. This should be introduced more consistently. - Alg 1: Is this algorithm implemented in pure Venturescript or as part of the engine? - fig 1: The figure does not match the definition: z_r should also be in a plate and the entire system should not be in a plate. - Sec 3.1: Where does the Naive Bayes assumption come from? Should z_r in the definition be a single variable (this would then also correspond with eq 2 and Fig 1). Minor comments: - Sec 2.1: This is a confusing section. - Sec 3.1: "learns learns"

Confidence in this Review

2-Confident (read it all; understood it all reasonably well)


Reviewer 3

Summary

The authors introduce conditional generative probabilistic models (CGPMs), which are essentially probabilistic models with a partially specified conditional dependence structure. A CPGM can embed submodels for which the conditional dependence structure need not be known, as along as it is possible to sample and evaluate the density for any conditional distribution on a subset of leaf node variables in the model. CGPMs can be composed into larger CGPMs, for which approximate implementations for the sampling and density functions can be defined using an importance sampling method. The authors present case studies of analysis performed using CGPMs that are implemented with BayesDB and Venturescript.

Qualitative Assessment

I think this is a strong paper. The authors on the one hand define a clear mathematical abstraction, the CPGM, and on the other hand show case studies of analysis performed with a probabilistic programming system that implements this abstraction. The CPGM is described in reasonably precise mathematical notation, which formalizes the aspects of this work that generalize beyond the specific implementation. The cases studies are not easy to follow without familiarity with the BayesDB system, but do a good job of illustrating the general idea to this approach, which is to provide a high level modeling/inference abstraction in which black box components can be embedded. The paper is reasonably clear. It is fairly heavy on notation, but most aspects of the presentation can be understood on a sufficiently close reading. I have pointed out point where presentation can be improved below. Questions and Suggestions - It would be helpful to explicitly define what is meant by "query" and "evidence" in the context of this paper, which currently has to be inferred by the reader from the simulate / logpdf interface. - What is the distinction between the observed output nodes D and the evidence node x_{r,E}? Does D refer values {x_r: r \in D} for which *all* variables are *always* observed? - In figure 1, why is there a plate around x and not around z? I would avoid using the the index i when writing {z_i : i \not= r} in the caption, since the authors use x_[r,i] in the next sentence. - I was unable to find a definition for {q_j} in the simulate interface. Presumably q_j \in [T]? Likewise the notation x_[r,Q] presumably refers to {x_[r,q] : q \in Q}? - What is meant by "the subset of the outputs x^a_[*, A]"? In particular, what does the "*" notation mean? - In algorithm 1, please define explicitly which symbol is bound to which input in the functions simulate and logpdf. Having to infer this from context makes it not obvious what is going on. - I'm confused by the simulate implementation. Line 3 generates a sample conditioned on both the evidence and the queried variables, does it not? So then, why would {x_[r,Q^k] \in s^m}?, since in line 24 we simulate \E^k? Should line 3 perhaps read Weighted-Sample(x[r,E_k]) - This is a matter of taste, but I would prefer for the authors to not use 'w' to describe a log weight, since most of the importance sampling literature uses 'w' to refer to an importance weight (i.e. the exponent of the log weight w used by the authors). - evdience -> evidence (please run a spell check)

Confidence in this Review

3-Expert (read the paper in detail, know the area, quite certain of my opinion)


Reviewer 4

Summary

The authors extend generative population models, a construct used in modeling tabular data for probabilistic inference, to be compositional. This allows for straightforward combinations of generative and discriminative methods.

Qualitative Assessment

This paper clearly presents an obviously good idea: making generative population models composable. This should be transparent and of interest to researchers with some knowledge of probabilistic programming and cross-categorization. I'm recommending an accept. Some suggestions for improvement: - In Figure 2, how is bayesdb:crosscat+OLS fewer lines than just OLS by itself? - The examples presented in Figures 2-4 are simple and clear but it seems worth adding an example that better shows off the ability of CGPMs to handle intra-row dependencies, e.g., modeling survey data where people are asked to rank a list of products. A couple of typos: - Line 143: you wrote that you composed Crosscat with random forests; did you mean to say clustering? - Figure 3c: "Magntiude" -------------- Update: I read the other reviews and rebuttal; my assessment is unchanged.

Confidence in this Review

3-Expert (read the paper in detail, know the area, quite certain of my opinion)


Reviewer 5

Summary

The authors present an abstraction over multiple probabilistic graphical models called "conditional generative probabilistic models" (CGPM). They show how one would express regression models using CGPMs and I can imagine many models would fit their computational model. They also show that CGPMs can be used to sample from and compute probabilities over probabilistic programs in VentureScript. They evaluate their model by predicting the lifetime of satellites and their deviations from Keplers Law.

Qualitative Assessment

Technical Quality The CGPM model does generalize many existing models but it would strengthen the paper if they can explicitly show these generalizations. Also it is not completely clear what these generalizations buy us beyond implementation convenience. Based on the experimental results, it seems that CGPMs can have lower error rates than other models by easily combining multiple models. Would that lead to newer interesting hybrid models ? Novelty/Impact While the CGPM is a novel generalization of existing models, it doesn't translate to any novel results about these kind of models. This might be something that could strengthen the paper. The satellite dataset is very interesting and it might be great to see if the authors have discovered some other interesting aspects. Clarity The paper is well written but would benefit from the description of the various model generalizations. Code snippets are not very helpful and could be dropped instead.

Confidence in this Review

2-Confident (read it all; understood it all reasonably well)